# Social Activity in Schizotypy: Measuring Frequency and Enjoyment of Social Events

**DOI:** 10.3390/bs14060474

**Published:** 2024-06-05

**Authors:** Candice C. DeBats, Danielle B. Abel, Morgan M. Sullivan, Sophia C. Koesterer, Imani S. Linton, Jessica L. Mickens, Madisen T. Russell, Lillian A. Hammer, Kyle S. Minor

**Affiliations:** 1Department of Psychology, Indiana University-Purdue University Indianapolis, Indianapolis, IN 46202, USA; cdebats@iu.edu (C.C.D.); danielle.abel@va.gov (D.B.A.); morgsull@iu.edu (M.M.S.); sophie-koesterer@uiowa.edu (S.C.K.); imlinton@iu.edu (I.S.L.); jmickens@iu.edu (J.L.M.); madtruss@iu.edu (M.T.R.); 2West Haven VA Medical Center, West Haven, CT 06516, USA; 3Department of Psychology, University of Indianapolis, Indianapolis, IN 46227, USA; 4Department of Psychology, University of Iowa, Iowa City, IA 52242, USA; 5Department of Psychology, University of Southern Mississippi, Hattiesburg, MS 39406, USA; lillian.hammer@usm.edu

**Keywords:** social functioning, social activities, schizotypy, negative traits

## Abstract

Improving social functioning deficits—a core characteristic of schizophrenia-spectrum disorders—is often listed by patients as a key recovery goal. Evidence suggests that social deficits also extend to people with schizotypy, a group at heightened risk for psychotic and other psychopathological disorders. One challenge of social functioning research in schizotypy is understanding whether social deficits arise from receiving less pleasure from social activities or from participating less in high-pleasure activities. However, limited information exists on what constitutes highly pleasurable, common social activities. In this study, 357 college students rated the frequency and enjoyment of 38 social activities. Our aims were to categorize activities based on their frequency and enjoyment, and whether these correlated with validated social functioning and schizotypy measures. We found that social activities could be characterized based on their frequency and enjoyment and created a frequency–enjoyment matrix that could be useful for future studies. Activities were correlated with social functioning, generally reaching a small effect size level, with increasing frequency and enjoyment showing associations with greater social functioning. Further, negative and disorganized—but not positive—traits were associated with less engagement and pleasure. Although follow-up studies in community samples are needed, our findings have the potential to help researchers and clinicians better understand which activities participants are more likely to engage in and derive pleasure from. The findings may also illustrate the extent to which social deficits may be due to less engagement or less pleasure from social activities, as well as which aspects of schizophrenia-spectrum disorders are associated with these facets of social functioning.

## 1. Introduction

Social functioning deficits, a hallmark of schizophrenia, are a critical focus of recovery efforts [1,2,3]. These deficits primarily relate to difficulties in forming and maintaining interpersonal relationships and effectively engaging in social interactions [4,5,6]. The relevance of social functioning extends beyond clinical definitions: it represents a fundamental aspect of human behavior, essential for quality of life and overall well-being [7,8]. Social deficits are not only prominent in those diagnosed with psychotic disorders but also extend to individuals with schizotypy [9,10,11]. Schizotypy is characterized by a range of traits that elevate the risk for psychotic and other psychopathological disorders [12,13]. Given that social functioning is vital to recovery in psychosis and schizotypy indicates elevated risk for these disorders, understanding social deficits may be important for identifying individuals who would benefit from early intervention strategies.

Research has indicated that schizotypy has different dimensions—positive (e.g., perceptual aberrations), negative (e.g., social anhedonia), and disorganized (e.g., odd speech) traits—and each is associated with varying degrees of social functioning impairment [13,14,15]. Notably, negative schizotypy shows the most substantial evidence of adversely affecting social connections, demonstrating associations with decreased social interest and engagement [9,16]. Positive and disorganized traits have shown less consistent associations but have been linked with unusual, eccentric social behavior [15] and diminished cognitive or affective empathy [17]. Understanding how schizotypy traits relate to aspects of social functioning, such as engagement and pleasure, may allow us to better conceptualize the underlying mechanisms of social deficits in schizotypy and, subsequently, guide the development of more personalized therapeutic approaches.

A central question is whether those high in schizotypy simply engage in fewer social activities, perceive these activities as less enjoyable, or if a combination of both factors reduces social functioning; see [18,19]. It is vital to distinguish between these possibilities and how they pertain to specific schizotypy traits. For example, if individuals with schizotypy participate less in social activities despite finding similar levels of anticipated pleasure as those low in schizotypy, this could indicate barriers in initiating or maintaining social connections. Conversely, if findings suggest lower anticipated pleasure, this might indicate a diminished desire for social interaction. Although the outcome—reduced social activity—is similar, understanding which explanation precipitates this reduction in social activity, and for which schizotypy traits, could lead to more effective intervention targets.

A current challenge in understanding whether people high in schizotypy engage less in social activities or choose activities that are less pleasurable is that it is difficult to discern which types of activities are frequently engaged in and the level of pleasure individual activities typically result in. To address these questions, our study’s primary objective is to characterize specific activities based on the frequency of occurrence and the enjoyment derived, thereby providing a more nuanced understanding of social engagement in this population. This approach allows for a detailed analysis of how social functioning and each type of schizotypal trait are associated with specific social activities. For instance, pronounced negative schizotypy traits might correlate with lower enjoyment of social activities, which could impact the motivation to participate and engage in social activities [20,21]. On the other hand, disorganized traits could lead to erratic or unpredictable engagement, impacting the quality and consistency of social interactions [22]. Insights gained from these analyses could allow more targeted interventions to effectively promote social engagement and reduce the chance of progression to more severe psychopathology.

We recognize that social functioning deficits in schizotypy likely stem from many factors, including but not limited to less engagement or pleasure in social situations. Reduced social skills, fewer prosocial behaviors, and difficulties in interpreting or responding to social cues are also significant contributors [10,23]. People high in schizotypy often struggle with understanding others’ perspectives and emotions [23]; this can lead to heightened cognitive demands, thereby exacerbating social difficulties [24,25]. This highlights the importance of collectively examining both social engagement and enjoyment across schizotypy traits. Additionally, the impact of schizotypy on deriving pleasure from social interactions is relatively underexplored. Research in schizophrenia has highlighted deficits in experiencing pleasure from social interactions [26,27,28]. Similar studies in schizotypy are scarce but show that those with negative schizotypy experience more negative affect in social situations [20] and may have difficulties anticipating and experiencing in-the-moment pleasure [21,29]. Few studies have examined how pleasure is impacted in social situations, however. Thus, our study aims to address this gap by investigating how high levels of positive, negative, and disorganized schizotypy are associated with perceptions of varied social contexts.

In order to clarify the nature of social deficits in schizotypy, our study’s primary objective was to create a 3 × 3 frequency–enjoyment matrix to characterize social activities into nine distinct categories based on their levels (high, moderate, low) of frequency and enjoyment. This will help identify which types of social activities are most and least prevalent and enjoyable for people with varying degrees of schizotypy traits. This distinction is important because while enjoyable activities are generally seen as beneficial to social functioning, less enjoyable activities (e.g., work interactions, obligatory social gatherings) also play a role in social functioning. A matrix will help us characterize activities into categories and parse out whether less interest in specific activities may be due their frequency, their enjoyment level, or a combination of both. In turn, this will aid in providing a better understanding of social functioning within schizotypy—and potentially schizophrenia—populations.

Additionally, we investigated how different types of social activities correlated with validated measures of social functioning and schizotypy traits. Our expectation was that the overall frequency and enjoyment of social activities would show positive associations, with a moderate effect or greater, with social functioning. We also hypothesized that inverse associations would be found between the overall frequency and enjoyment of social activities with negative traits (moderate effect or greater), positive traits (small effect), and disorganized traits (small effect). If these hypotheses are supported, this would provide evidence for the explanation that those with schizotypy may engage in fewer social activities because they derive less pleasure from them. If our hypotheses are not supported, this could suggest several possibilities. For example, if frequency is inversely associated with schizotypy traits but enjoyment is the same, this would suggest that other factors are the reason why social activities are pursued less frequently by those higher in schizotypy. No specific hypotheses were made regarding exploratory analyses of how different types of activities (i.e., cells) within the frequency–enjoyment matrix are related to our other constructs of interest (e.g., are high-frequency, low-enjoyment activities associated with positive schizotypy traits?). Our approach of examining specific traits is relevant given the heterogeneity of schizotypy and the varied ways it can influence social behavior and experiences. By examining these correlations, we hope to shed light on the complex interplay between schizotypy traits and social functioning, offering insights that could inform more tailored and effective therapeutic interventions.

## 2. Methods

### 2.1. Participants

The study consisted of 357 undergraduate students recruited from Indiana University Purdue University–Indianapolis (IUPUI; *n* = 243) and the University of Southern Mississippi (*n* = 114). Students were recruited from January 2021 through December 2022 using Sona systems (https://www.sona-systems.com/), an online research pool used by the Psychology Departments at IUPUI and USM. The eligibility criteria included (1) no history of psychotic symptoms; (2) age 18–60 years old; (3) English fluency; (4) ability to give informed consent; and (5) passed a validity screen consisting of four questions from the Chapman Infrequency Scale [30]. The study was conducted entirely online in one continuous session that lasted approximately one hour. All participants were compensated for their time with either research credits or USD 10 for participation.

### 2.2. Measures

**Social Activities:** To assess social activities, we used items from the Pleasant Activities List (PAL [31,32]). For this study, we only selected items that represented social activities given that this was the focus of the study. Items from the PAL measure both the frequency and enjoyment that one experiences while completing a certain task or activity. They ask participants to rate activities based on the past 30 days from when the questionnaire was administered. The social activities items from the PAL consisted of 38 activities, and each activity was separated into “frequency” and “enjoyability” sections, which were then answered on a five-point scale ranging from “not at all” to “very much”. The original scale contains subscales and we implemented at least one question from six subscales (social activities; miscellaneous activities; intimacy or personal attention; culture, science, and/or traveling; passive or relaxing activities; and sport-related activities). For this study, we summarized frequency and enjoyability for the total scale and individual items. Individual items were then examined to determine if groupings could be made based on their frequency and enjoyment.

**Social Functioning:** The Social Adjustment Scale Self-Report (SAS-SR [33]) was used to measure social functioning. The SAS-SR consists of 14 items, and each item is scored on a five-point scale representing normal to maladaptive social performance. The primary goal of the 14-item SAS-SR is to allow for shorter administration while covering the primary areas from the 54-item SAS-SR [33]. The shortening of this scale does not hinder validity, making the SAS-SR useful for epidemiologic studies and studies requiring a brief social functioning measure.

**Schizotypy:** The Multidimensional Schizotypy Scale-Brief (MSS-B [34]) was used to assess schizotypy in this study. Self-report questionnaires, such as the MSS-B, are commonly implemented in college students to assess schizotypy traits [11,12,13,29]. It consists of 38 true–false questions, which are used to calculate positive, negative, or disorganized dimensions. The MSS-B maintains content coverage and has exhibited good reliability and validity in assessing associations between schizotypy dimensions and impairment [14]. Using the MSS-B provides us with a useful tool to study risk, resilience, and the development of schizotypy [14].

### 2.3. Proposed Analyses

Analyses were completed in four parts. First, we calculated demographic data for the full sample. Second, we computed descriptive data on the 38 social activities assessed and created a 3 × 3 frequency–enjoyment matrix (i.e., high/moderate/low for frequency and enjoyment) based on average frequency and enjoyment reported. Third, we ran a series of Spearman’s correlations examining relationships between overall social activities, social activity matrix categories (e.g., high frequency/high enjoyment), and social functioning. Finally, we ran Spearman’s correlations examining relationships between overall social activities and social activity matrix categories with positive, negative, and disorganized schizotypy traits. Spearman’s correlations were selected in our analyses based on our variables of interest showing a non-normal distribution when Shapiro–Wilk’s tests were performed (all *p*-value’s < 0.05). For this study, all significance values were set at *p* < 0.01. A more conservative estimate was used due to the large number of correlations conducted.

## 3. Results

### 3.1. Demographic Data

Our study primarily consisted of young adults, with an average age of 20.71 (SD = 5.18) years old. Two-thirds of the sample (68.2%) were female, with less than a third reporting their gender as male (29.6%) or non-binary (2.3%). Ethnicity for the sample was 89.3% non-Hispanic, 7.6% Hispanic or Latino, and 3.1% unknown. The highest self-reported race, which was assessed separately from ethnicity, was White (58.0%), followed by Black or African American (23.7%), multi-racial (6.2%), Southeast Asian descent (5.4%), or other (4.5%). Less than 1% of participants reported their race as American Indian or Alaskan Native or Native Hawaiian or Other Pacific Islander. Thirty percent of participants reported past therapy sessions, with 10% attending 25 or more sessions, highlighting variance in previous experience with mental health services.

### 3.2. Separating Social Activities Based on Frequency and Enjoyment

Participants averaged scores of 2.50 (0.55) for frequency and 3.13 (0.61) for enjoyment of social activities, indicating moderate levels of both. To create the frequency–enjoyment matrix, we defined ‘high’ scores to be above the mean and above the midpoint on the scale (3) for frequency or enjoyment (i.e., >3.00 for frequency, >3.50 for enjoyment). ‘Moderate’ scores were defined as scores close to the mean and inclusive of the midpoint on the scale (i.e., 2.01–3.00 for frequency, 2.51–3.50 for enjoyment). ‘Low’ scores were defined as scores below the mean and the midpoint on the scale (i.e., ≤2.00 for frequency, ≤2.50 for enjoyment).

We used these criteria to categorize social activities into a matrix based on frequency and enjoyment (see Figure 1), providing a structured overview of common activity patterns. As Figure 1 shows, at least one social activity was observed for eight of the nine categories, with ‘high frequency/low enjoyment’ being the lone exception. The most activities (13) were observed for the ‘moderate frequency/moderate enjoyment’ category. Based on our criteria, scores were summed and average scores were calculated for each of the represented subcategories based on frequency and enjoyment. Table 1 lists means and standard deviations for each individual activity.

### 3.3. Relationships between Social Activities and Social Functioning

Social functioning scores varied from 1.25 to 9.88, with an average score of 3.25 (SD = 1.20). We observed small effect size correlations between social activities and social functioning. Total average frequency and enjoyment scores were correlated with social functioning, with scores of *ρ*(355) = 0.16, *p* = 0.003 and *ρ*(355) = 0.16, *p* = 0.002, respectively. As shown in Table 2, the strength of these associations varied based on the frequency and enjoyment of the activities measured. The strongest relationship with frequency was observed for moderate-frequency, low-enjoyment activities; the strongest relationship with enjoyment was found for high-frequency, high-enjoyment activities.

### 3.4. Relationships between Social Activities and Schizotypy Traits

The mean scores for the schizotypy scales administered in this study were 2.26 (SD = 2.43) for positive traits (range: 0–13), 2.21 (SD = 2.32) for negative traits (range: 0–13), and 2.24 (SD = 2.87) for disorganized traits (range: 0–12). Associations between social activities and schizotypy differed based on the type of schizotypy trait assessed. Total average frequency and enjoyment scores were not correlated with positive traits, with scores of *ρ*(355) = 0.09, *p* = 0.111 and *ρ*(355) = 0.00, *p* = 0.993. We observed medium inverse correlations between negative traits and both frequency, *ρ*(355) = −0.34, *p* < 0.001, and enjoyment scores, *ρ*(355) = −0.31, *p* < 0.001. Associations between social activities and disorganized traits were smaller yet still reached the level of significance, with scores of *ρ*(355) = −0.18, *p* < 0.001, and *ρ*(355) = −0.18, *p* < 0.001.

Relationships between schizotypy traits and social functioning were also assessed. Social functioning was associated with positive traits, *ρ*(355) = −0.25, *p* < 0.001; negative traits, *ρ*(355) = −0.30, *p* < 0.001; and disorganized traits, *ρ*(355) = −0.38, *p* < 0.001. These associations were slightly weaker than those observed for overall social activities with negative traits but stronger for positive and disorganized traits.

As shown in Table 3, the strength of associations between schizotypy traits and social activities varied based on the frequency and enjoyment of the activities measured. Positive traits did not generally reach the level of significance for frequency or enjoyment. Negative traits demonstrated the strongest relationships when frequency and enjoyment were high; relationships appeared to become weaker as frequency and enjoyment decreased. Disorganized traits exhibited several small effects with no clear pattern in terms of frequency and enjoyment.

## 4. Discussion

This study focused on examining the nature of social deficits in schizotypy by exploring relationships between schizotypy traits and specific social activities. Three findings of interest emerged. First, we were able to characterize social activities in college students based on their level of frequency and the enjoyment derived. At least one activity fit into eight of the nine categories in our frequency–enjoyment matrix. Second, social activities were associated with social functioning, although the observed relationships were generally small. Third, social activities were inversely associated with negative and disorganized—but not positive—schizotypy traits. Unsurprisingly, negative traits were the most strongly related to both the frequency and enjoyment of social activities. These findings provide a nuanced overview of social activity patterns in both college students and those exhibiting different types of schizotypy traits.

The frequency–enjoyment matrix we developed to categorize social activities serves as a key contribution from this study. This framework can be applied to better understand whether those high in schizotypy derive less enjoyment from social activities or if they simply engage in less enjoyable social pursuits. Based on our findings, it does appear that enjoyment is more strongly associated with negative schizotypy traits for activities in the high-enjoyment cell of the matrix; in the low-enjoyment cells, these relationships tend to be weaker. Frequency followed a similar pattern, meaning that frequency and enjoyment are not as strongly related to negative traits in lower-propensity, more obligatory activities but these relationships become more apparent for more frequent, enjoyable activities. This provides evidence that some combination of receiving less enjoyment and also pursuing enjoyable activities less frequently may help explain social deficits—at least for those high in negative schizotypy. Future work should utilize our frequency–enjoyment matrix within other methodologies. By integrating methodologies such as ecological momentary assessment (EMA), this framework could yield important insights into the types of activities participants pursue. For example, a common question in EMA research is how to compare different types of social activities across participants (e.g., is a phone call the same level of social interaction as sharing a meal with friends?). Findings from this study can serve as a guide to determine normative levels of enjoyment and frequency for common activities. This will allow researchers to better study functioning in daily life.

In this study, we observed that activities with moderate frequency and enjoyment were most prevalent. Generally, the number of social activities in a given category tended to increase when enjoyment—but not frequency—was higher. This highlights the need to consider the quality of social activities, not just their quantity, as a way to enhance social functioning [22,35]. This framework also has practical implications for clinicians: it could serve as a guide for suggesting activities that clients are more likely to enjoy and are relatively easy to do when using techniques like behavioral activation (e.g., visiting family, calling a friend). Further, although we focused primarily on schizotypy, the approach used here could also translate to other conditions characterized by social deficits, such as autism spectrum disorder or social anxiety disorder [36,37,38,39,40]. This has the potential to provide a more comprehensive understanding of social functioning across clinical conditions.

The relatively small association between overall social activities and social functioning was somewhat surprising, as we expected these relationships to be more robust. There were not clear patterns across high-, moderate-, or low-frequency cells; the strongest relationships were observed in the high-frequency, high-enjoyment and moderate-frequency, low-enjoyment cells. Our findings also suggest that although social activities are integral to social relationships, their frequency and enjoyment may represent distinct dimensions from the traditionally rated aspects of social functioning, such as forming and maintaining interpersonal relationships [5]. The differential relationships that social activities showed with schizotypy traits compared to social functioning supports this possibility. However, social activities are a way that social bonds are often strengthened, and the activities listed here cover a wide range of common experiences. Future studies in clinical populations should investigate whether these associations differ from those found here in a college sample.

Schizotypy traits demonstrated unique patterns in their relationship with social activity frequency and engagement. Notably, medium-sized associations between negative schizotypy traits and reduced engagement and pleasure in social activities were observed. This finding aligns with the broader literature, indicating that negative traits are often associated with diminished social behavior across many domains [9,41,42,43]. It also aligns with the wider schizophrenia literature that suggests those with negative symptoms have less motivation to pursue social activities and may receive less pleasure from them when they do participate [44,45]. In contrast, the weaker associations observed between disorganized traits and the absence of significant associations between positive traits and social activity frequency and enjoyment suggest diverse pathways of social dysfunction. This might indicate that the eccentric behavior often linked with positive and disorganized schizotypy does not necessarily hinder the frequency or enjoyment of social interactions. It is possible that individuals with predominantly positive schizotypy traits, in particular, retain their interest and pleasure in social activities, but potentially engage in less conventional ways. This distinction is important for clinicians and researchers to disentangle, as it underscores the need for varied therapeutic approaches targeting different facets of schizotypy.

The differential associations observed suggest targeted interventions may be appropriate for those with distinct schizotypy traits. For individuals with pronounced negative traits, interventions might emphasize enhancing motivation and pleasure in social activities. Techniques such as behavioral activation, which has shown efficacy in increasing engagement in pleasurable activities [46,47], could be adapted for this population. Similarly, cognitive behavioral therapy that addresses negative beliefs about social interactions and teaches social skills might also be beneficial. Metacognitive therapy, which has been effective at improving cognitive symptoms and social functioning in schizophrenia, could also be adapted for this purpose [48]. For those with positive and disorganized traits, the focus might shift to improving the quality and appropriateness of their social interactions rather than merely increasing their frequency. Incorporating elements of social skills training could be one way to provide these individuals with practical tools to navigate social situations more effectively [49,50,51].

This study’s strength lies in its nuanced evaluation of social activities and their associations with schizotypy traits. However, limitations also exist. This study relies on self-report ratings to assess the frequency and enjoyment of social activities. Follow-up studies are needed to assess whether these ratings change over time and how closely they overlap with ratings on ambulatory assessments. Second, the study only examined social activities at one time point and did not examine the quality of the interactions occurring in these activities; see [22,34,52]. Exploring temporal relationships and the types of interactions people engage in during social activities could provide additional information on whether lower scores represent a lack of interest or lower engagement in social situations. Related to this, a third limitation is that social activities were not compared to non-social activities. Future studies should do so to observe whether there are discrepancies in frequency or enjoyment scores. Fourth, this study did not explore potential underlying mechanisms driving reduced social engagement and pleasure. Examining mechanisms such as cognitive deficits, depression, or childhood trauma could provide additional insights into addressing social deficits [53,54,55].

In sum, our study provides insights into the complex interplay between schizotypy traits and social functioning. We created a frequency–enjoyment matrix that can be applied to ecological momentary assessment and laboratory studies examining schizotypy. Within this matrix, we observed that social activities exhibited a small positive association with social functioning and a small negative association with disorganized traits. Inverse associations between frequency, enjoyment, and negative traits were more robust. Understanding these relationships is important and may allow for the development of targeted interventions to improve social outcomes in people high in different types of schizotypy traits. Future research should continue to explore the complex relationships between schizotypy, social engagement, and enjoyment, expanding upon our findings to develop a more comprehensive understanding of social functioning across the schizophrenia spectrum.

## Figures and Tables

**Figure 1 behavsci-14-00474-f001:**
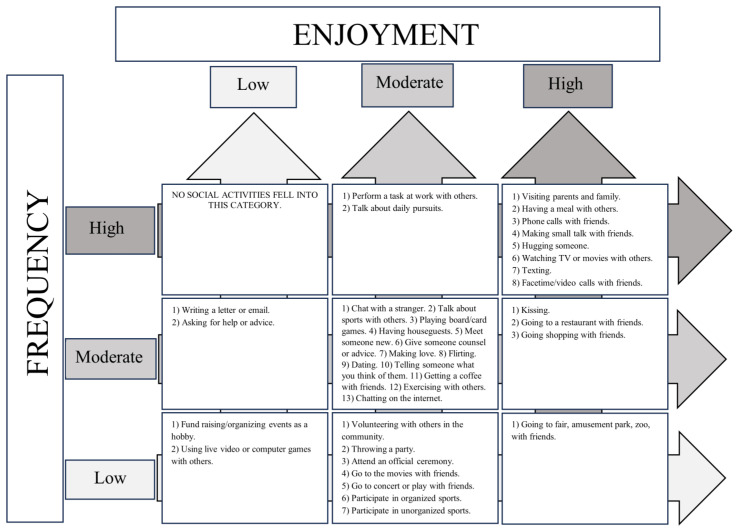
The frequency–enjoyment matrix of social activities.

**Table 1 behavsci-14-00474-t001:** Means and standard deviations for sample (*n* = 357) across the frequency–enjoyment matrix.

		High Enjoyment			Moderate Enjoyment			Low Enjoyment	
	*n*	Frequency	Enjoyment	*n*	Frequency	Enjoyment	*n*	Frequency	Enjoyment
High Frequency	8	3.39 (0.73)	3.87 (0.66)	2	3.19 (1.01)	3.15 (0.95)	0	N/A	N/A
Moderate Frequency	3	2.70 (1.04)	3.83 (0.99)	13	2.43 (0.65)	3.17 (0.72)	2	2.57 (0.92)	2.27 (0.89)
Low Frequency	1	1.77 (1.07)	3.59 (1.56)	7	1.70 (0.68)	2.87 (1.00)	2	1.73 (0.86)	2.34 (1.07)

Notes. *n* represents the number of social activities per category. N/A is not applicable.

**Table 2 behavsci-14-00474-t002:** Correlations between social functioning and social activities for the sample (*n* = 357) across the frequency–enjoyment matrix.

	High Enjoyment		Moderate Enjoyment		Low Enjoyment	
	Frequency	Enjoyment	Frequency	Enjoyment	Frequency	Enjoyment
High Frequency	0.16 *	0.20 **	−0.11	0.09	N/A	N/A
Moderate Frequency	0.06	0.02	0.08	0.12	0.20 **	0.17 *
Low Frequency	0.07	−0.01	0.20 **	0.13	0.14 *	0.10

Notes. * *p* < 0.01; ** *p* < 0.001. N/A is not applicable.

**Table 3 behavsci-14-00474-t003:** Correlations between social activities and schizotypy traits for the sample (*n* = 357) across the frequency–enjoyment matrix.

	High Enjoyment		Moderate Enjoyment		Low Enjoyment		
	Frequency	Enjoyment	Frequency	Enjoyment	Frequency	Enjoyment	
High Frequency	0.04	−0.06	−0.03	−0.09	N/A	N/A	**Positive Schizotypy Traits**
Moderate Frequency	0.08	0.10	0.11	0.04	0.01	−0.05	
Low Frequency	0.01	0.03	0.04	−0.01	0.08	0.05	
High Frequency	−0.36 **	−0.35 **	−0.12	−0.21 **	N/A	N/A	**Negative Schizotypy Traits**
Moderate Frequency	−0.28 **	−0.21 **	−0.29 **	−0.29 **	−0.16 *	−0.18 **	
Low Frequency	−0.10	−0.07	−0.17 *	−0.17 **	−0.06	−0.10	
High Frequency	−0.16 *	−0.16 *	−0.03	−0.14 *	N/A	N/A	**Disorganized Schizotypy Traits**
Moderate Frequency	−0.14 *	−0.04	−0.13	−0.16 *	−0.10	−0.18 **	
Low Frequency	−0.11	−0.10	−0.15 *	−0.15 *	−0.01	−0.03	

Notes. * *p* < 0.01; ** *p* < 0.001. N/A is not applicable.

## Data Availability

Data for this project are available upon request.

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
