# Peer review of "Social Activity in Schizotypy: Measuring Frequency and Enjoyment of Social Events"

_behavsci, 2024, doi:10.3390/bs14060474_

Round 1
Reviewer 1 Report (Previous Reviewer 3)
Comments and Suggestions for Authors
I thank the authors for the consideration given to my comments, and I have no further comments to make.
Reviewer 2 Report (Previous Reviewer 2)
Comments and Suggestions for Authors
The manuscript has been improved and the current version is clear and easy to read.
This manuscript is a resubmission of an earlier submission. The following is a list of the peer review reports and author responses from that submission.
Round 1
Reviewer 1 Report
Comments and Suggestions for Authors
Review report attached

vague and lacks rationale
Reviewer 2 Report
Comments and Suggestions for Authors
It is really a pleasure to read the manuscript, it is well organized and well written. Using a frequency-enjoyment matrix, the study examined the associations between social activities and schizotypal symptoms and social functioning in a sample of undergraduate students. The results showed that negative and disorganized traits - but not positive traits - were associated with less engagement and enjoyment. The manuscript presents a well-designed study that makes a valuable contribution to the field, examining the associations between social activities and schizotypal symptoms is an important and timely topic that deserves further investigation.
I just have a few minor comments as follows:
1. Which correlation analysis do you use, Pearson or Spearman? Have you checked the raw data distribution?
2. Tables 2 and 3 report a large number of correlations, which increases the risk of false positive errors (Type I errors) due to the multiple comparisons being made. This is especially concerning given the relatively small sample sizes in each category, as shown in Table 1. Smaller sample sizes tend to result in lower statistical power, further exacerbating the multiple comparisons problem. To address this issue, the authors should consider applying an appropriate multiple comparison correction method, such as the Bonferroni correction or the false discovery rate (FDR) control, to adjust the p-values and control the family-wise error rate or the false discovery rate.
3. Age and gender may influence the results and their associations with social activities, social functioning and schizotypy could be explored.
Reviewer 3 Report
Comments and Suggestions for Authors
The authors sought to establish the relationship between schizotypal traits in a non-clinical population and the frequency of enjoyment of 38 pleasant social activities, alongside social functioning. Specifically, they were interested in whether dimensions of schizotypy would differentially correlate with frequency and or enjoyment of social activities to ascertain whether social difficulties are due to less enjoyment or other obstacles. To this end they had participants take a scale of schizotypy traits, a scale of social functioning, and rate 38 social activities in terms of their frequency and enjoyment. The social activities were categorised in a matrix of high/moderate/low enjoyment and frequency. The results showed that social functioning correlates positively with frequency and enjoyment of social functioning. Frequency and enjoyment of social interactions correlated inversely with the negative schizotypy dimension, and to a lesser extent with the disorganised schizotypy dimension. The negative schizotypy dimension correlated inversely with
Overall, this was a clearly written and well organised manuscript. The design is straightforward and to the point. The results are clearly summarised and the discussion has relevant implications. This paper would make a contribution to the literature, pending some revisions (below) that I think can be easily addressed by the authors.
Introduction
“Research has indicated that schizotypy has different dimensions—positive, negative, and disorganized traits—and each is associated with varying degrees of social functioning impairment”
Please could the authors add more description about these different dimensions, maybe with examples.
“Research in schizophrenia has highlighted deficits in experiencing pleasure from social interactions [26-28] but similar studies in schizotypy are scarce [but see 20,21,29].”
The references 20,21,29 seem relevant here and deserve to be reviewed, even if they are only tangentially related. As a reader it is unclear why we are referred to these studies, especially after reading that studies are scarce.
“…our study’s primary objective was to create a 3 × 3 frequency-enjoyment matrix…”
At this point it is unclear what the 3 x 3 is referring to. Only later it becomes clear it refers to high/moderate/low. Either introduce these categories here or remove the reference to “3 x 3” (so that it just states “to create a frequency-enjoyment matrix”).
The hypotheses could be tested by correlating schizotypy dimensions with overall frequency and enjoyment scores, without the matrix. I think more justification for creating the matrix is therefore required. I agree with the authors that it adds more nuance, but some guidance on how this nuance is to be interpreted is required. For example, whilst enjoyable social activities are important for social functioning, unpleasant social interactions (such as work interactions) are also important. What does it mean if schizotypy dimensions or social functioning correlate with frequency/enjoyment in different cells of the matrix?
“Our expectation was that overall frequency and enjoyment of social activities would show positive associations, at a moderate effect or greater, with social functioning. We also hypothesized that inverse associations would be found between overall frequency and enjoyment of social activities with negative traits (moderate effect or greater), positive traits (small effect), and disorganized traits (small effect).”
The initial justification for the study is to disentangle the contributions of enjoyment or other barriers to social activities. That is, if social activities are reduced in those high in schizotypy because they enjoy them less, then there would be a negative association between schizotypy and both frequency and enjoyment. If social activities are reduced because of other barriers (unrelated to enjoyment), then there would be a negative correlation between schizotypy and frequency only, and not with enjoyment. The hypotheses presented at the end of the introduction already seem to presume the former and not the latter. Could the authors please clarify.
“No specific hypotheses were made regarding exploratory analyses of different types of activities within the frequency-enjoyment matrix and other constructs of interest.”
Could the authors please clarify what this means. Even if you don’t have hypotheses, it would still be helpful to know what analyses you will conduct, and what the other constructs of interest are.
Method
“…we used items from the Pleasant Activities List (PAL [31])”
Could the authors provide a more suitable reference for the PAL, as it is not clear how to find the original from the information given in the reference list. The only one I could find was (which looks to be a different version): Roozen HG, Wiersema H, Strietman M, Feij JA, Lewinsohn PM, Meyers RJ, Koks M, Vingerhoets JJ. Development and psychometric evaluation of the pleasant activities list. Am J Addict. 2008 Sep-Oct;17(5):422-35. doi: 10.1080/10550490802268678. PMID: 18770086.
“For this study, we only selected items that represented social activities given that this was the focus of the study”.
To my mind, this is one of the biggest drawbacks of the study design. The study could add even more nuance to our understanding if it could show different relationships between schizotypy dimensions and social and non-social frequency/enjoyment. The inclusion of non-social activities could also tell us if a reduction in social activities is due to a reduced enjoyment of social interactions specifically or the result of a general reduction in taking pleasure from activities in general (anhedonia is a negative feature of schizophrenia as well, which is why a more detailed description of these negative symptoms is required, as mentioned above). The authors should justify more why they only chose social activities.
Results
“To create the 3 × 3 frequency-enjoyment matrix, we defined ‘high’ scores to be above the mean and above moderate (3) frequency or enjoyment (i.e., > 3.00 for frequency, > 3.50 for enjoyment). ‘Moderate’ scores were defined as scores close to the mean and inclusive of moderate on the scale…”
The duel use of the word ‘moderate’ in consecutive sentences leads to a bit of confusion. In the first it refers to the mid point of the rating scale. In the second, it refers to the category of the matrix. Could the first not be changed to “mid-rated (3)”?
General comment: Did the authors apply a correction for multiple comparisons when correlating based on the different cells of the matrix? If a stricter alpha level is applied, the pattern of results should become clearer in Table 3.
Discussion
“This framework can be applied to better understand whether those high in schizotypy derive less enjoyment from social activities or if they simply engage in less enjoyable social pursuits.”
Could the authors provide a stronger conclusion in this regard. As mentioned above, discerning the relative contributions seemed to be the rationale for the study, but was not a hypotheses – could the authors conclude either way from the current data?
“This suggests that although social activities are integral to social relationships, their frequency and enjoyment may represent distinct dimensions from the traditionally rated aspects of social functioning, such as forming and maintaining interpersonal relationships”
Again, the issue of engaging successfully with unpleasant or less enjoyable social interactions being important for social functioning could be addressed here.
“Third, this study did not explore potential underlying mechanisms driving reduced social engagement and pleasure. Examining mechanisms such as cognitive deficits, depression, or childhood trauma could provide additional insights into addressing social deficit”
This limitation refers back to the lack of inclusion of non-social activities, which may shed some light on whether negative schizotypy traits are associated with enjoyment of social activities specifically or enjoyment of activities in general. This underscores the need for a clearer justification for only including social activities.
